

# Predicting central cervical lymph node metastasis in papillary thyroid microcarcinoma using deep learning

Yu Wang[1], Hai-Long Tan[1], Sai-Li Duan[1], Ning Li[1], Lei Ai[1] and Shi Chang[1,2,3]

[1] Department of General Surgery, Xiangya Hospital, Central South University, Changsha, Hunan, China
[2] Clinical Research Center for Thyroid Disease in Hunan Province, Changsha, Hunan, China
[3] Hunan Provincial Engineering Research Center for Thyroid and Related Diseases Treatment Technology, Changsha, Hunan, China

## ABSTRACT

**Background**. The aim of this study is to design a deep learning (DL) model to preoperatively predict the occurrence of central lymph node metastasis (CLNM) in patients with papillary thyroid microcarcinoma (PTMC).

**Methods**. This research collected preoperative ultrasound (US) images and clinical factors of 611 PTMC patients. The clinical factors were analyzed using multivariate regression. Then, a DL model based on US images and clinical factors was developed to preoperatively predict CLNM. The model's efficacy was evaluated using the receiver operating characteristic (ROC) curve, along with accuracy, sensitivity, specificity, and the $F_1$ score.

**Results**. The multivariate analysis indicated an independent correlation factors including age $\geq$55 (OR = 0.309, $p < 0.001$), tumor diameter (OR = 2.551, $p = 0.010$), macrocalcifications (OR = 1.832, $p = 0.002$), and capsular invasion (OR = 1.977, $p = 0.005$). The suggested DL model utilized US images achieved an average area under the curve (AUC) of 0.65, slightly outperforming the model that employed traditional clinical factors (AUC = 0.64). Nevertheless, the model that incorporated both of them did not enhance prediction accuracy (AUC = 0.63).

**Conclusions**. The suggested approach offers a reference for the treatment and supervision of PTMC. Among three models used in this study, the deep model relied generally more on image modalities than the data modality of clinic records when making the predictions.

## INTRODUCTION

The occurrence of thyroid nodules is prevalent and there has been a steady increase in thyroid cancer cases over the past few decades (*Ferlay et al., 2019*). This is mainly due to the improvement in imaging techniques for screening, such as ultrasonography to increase the papillary thyroid carcinoma (PTC) detection rate, especially small ones (*Du et al., 2018*). Papillary thyroid microcarcinoma (PTMC) is defined as a PTC with a maximum diameter of 10 mm or smaller (*Haugen et al., 2016*). Some studies illustrated that the postoperative

Corresponding author
Shi Chang, changshi@csu.edu.cn

outcomes of PTMC without clinically evident extrathyroid extension (ETE) or lymph node metastases (LNM) are extremely favorable (*Ito et al., 2012*; *Yu et al., 2011*). Thus, they are called low-risk PTMC. Active surveillance rather than immediate surgery of low-risk PTMC is receiving increasing attention (*Sugitani, 2023*).

Although PTC is regarded as an indolent tumor, a portion of cancer cells will metastasize to lymph nodes around the thyroid gland. LNM includes central and lateral LNM (CLNM and LLNM, respectively) and it often first manifests in the central region and then the lateral region (*Haugen et al., 2016*). The presence of LNM is deemed a crucial indicator for forecasting PTC prognosis, deciding the surgical method, and is seen as a substantial risk factor for patients' high recurrence rate (*Yu et al., 2020*). Therefore, it is recommended that ultrasound (US) evaluations of the cervical lymph nodes are performed for all individuals with confirmed or potential thyroid nodules (*Haugen et al., 2016*). Preoperative ultrasound is a valuable tool in assessing LLNM in patients with PTC and can provide relatively reliable information of the lateral neck to assist in surgical management. However, the identification of CLNM by ultrasound has encountered significant challenges, since preoperative ultrasound can only detect 20–31% of CLNM (*O'Connell et al., 2013*). An efficient method to anticipate the risk of CLNM before surgery and to direct the clinical diagnosis and treatment process is urgently needed.

To address this problem, earlier studies attempted to employ clinical factor-based statistical methods to construct analysis models for LNM predictions in PTC patients (*Feng et al., 2022a*; *Wang et al., 2023*). With the development of technology, radiomics has attracted much attention in the precise diagnosis. Many studies extracted high-throughput radiomic features (HTRF) of the US images and established the relationship between these HTRF and LNM status (*Jiang et al., 2020*; *Shi et al., 2022*). Although the above studies have shown that the features of PTC lesion in US images were highly correlated with LNM, the prediction performance was not ideal.

Recently, deep learning (DL) algorithms have attracted considerable interest because of their exceptional performance in tasks related to image recognition. DL algorithms assisting medical diagnosis based on CT and MRI already had a wide range of applications (*Bandyk et al., 2021*). Although the application of DL algorithms in US images has achieved some outcomes (*Yu et al., 2020*), it is still in the early stage of clinical trials. In this study, we developed a DL model using US images and clinical factors of thyroid lesions. The model assists in generating a pre-surgical forecast of the risk of CLNM in patients suffering from papillary thyroid microcarcinoma and improving the effectiveness of patients' management.

## MATERIALS & METHODS

### Patients

This retrospective study was approved by the Ethics Committee of the Institutional Review Board of Xiangya Hospital, Central South University (202211733). Due to the removal of all patient identifying information, there was no need for informed consent.

The PTMC cases were selected from a group of 2,302 patients who underwent either thyroid lobectomy or total thyroidectomy along with cervical lymph node dissection

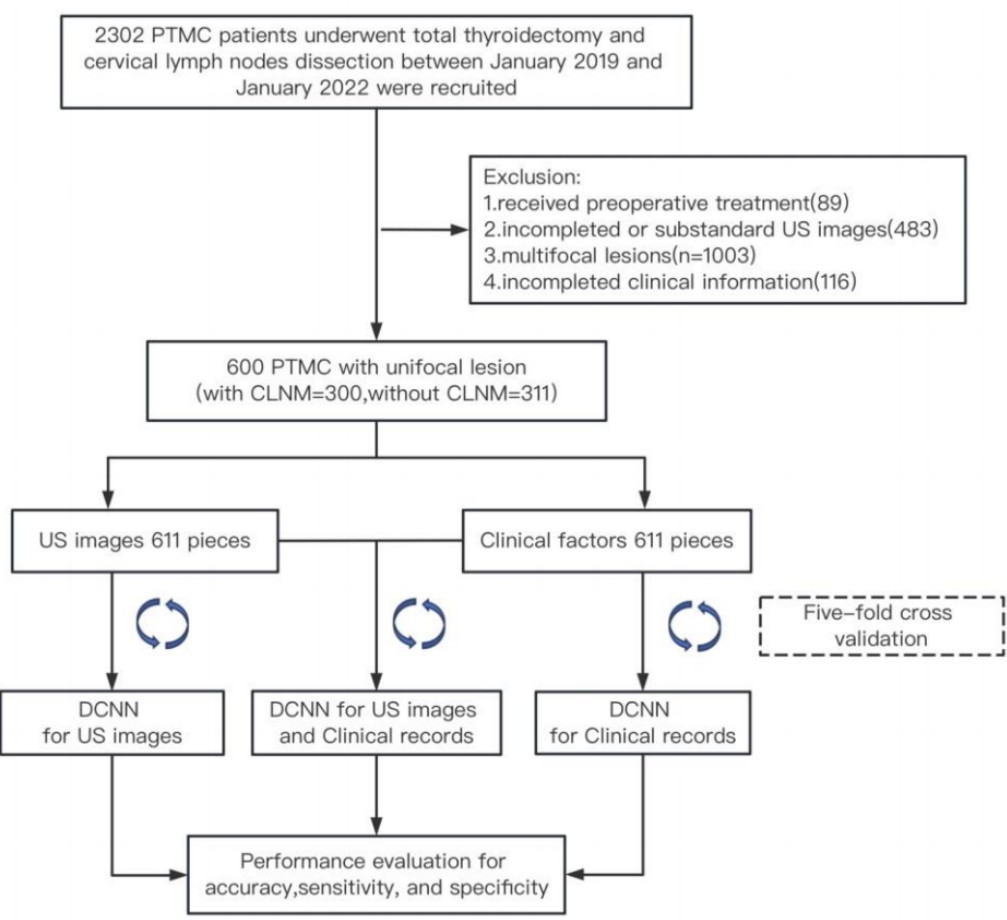

**Figure 1** **The data screening process.** PTMC, papillary thyroid microcarcinoma; CLNM, central lymph node metastasis; DCNN, deep convolutional neural network; US, ultrasound.

(LND) between January 2019 and January 2022. Figure 1 illustrates the data screening methodology. The criterion for inclusion included: (1) patients received a thyroid US diagnosis with available US images; (2) pathological confirmation of patients having PTMC; and (3) patients had a pathological lymph node diagnosis after cervical LND. The criteria for exclusion included the following: (1) patients who underwent treatment prior to their operation; (2) insufficient or substandard ultrasound images, such as the nodules were excessively large for full image capture, or there were measuring lines present in the ultrasound images; (3) patients diagnosed with multifocal lesions; (4) patients whose clinical information was incomplete.

Finally, a total of 611 PTMC cases with unifocal lesions were involved, including 300 metastatic cases and 311 non-metastatic cases. All clinical factors (CFs) were collected from the hospital information system. A summary of patient demographics, the classification of the Thyroid Imaging Reporting and Data System (TI-RADS), and clinicopathological features is provided in Table 1.

**Table 1** **The clinicopathological characteristics of PTMC patients with CLNM and without CLNM.**

| Characteristics | Total 611(%) | CLNM(-) 311(%) | CLNM(+) 300(%) | P |
|---|---|---|---|---|
| Gender | | | | |
|     male | 155(25.4%) | 66(21.2%) | 89(29.7%) | **0.016**[a] |
|     female | 456(74.6%) | 245(78.8%) | 211(70.3%) | |
| Age (mean ± SD, years) | 40.68 ± 10.79 | 43.29 ± 11.00 | 37.89 ± 9.89 | **<0.001**[b] |
|     <55 | 540(88.4%) | 260(83.6%) | 280(93.3%) | **< 0.001**[a] |
|     ≥55 | 71(11.6%) | 51(16.4%) | 20(6.7%) | |
| Tumor diameter (mean ±SD, cm) | 0.62 ± 0.24 | 0.58 ± 0.24 | 0.65 ± 0.24 | **< 0.001**[b] |
| Hashimoto's thyroiditis | | | | |
|     Yes | 100(16.4%) | 49(15.8%) | 51(17.0%) | 0.678[a] |
|     No | 511(83.6%) | 262(84.2%) | 249(83.0%) | |
| Shape | | | | |
|     regular | 100(16.4%) | 49(15.8%) | 51(17.0%) | 0.678[a] |
|     irregular | 511(83.6%) | 262(84.2%) | 249(83.0%) | |
| aspect ratio | | | | |
|     <1 | 388(63.5%) | 187(60.1%) | 201(67.0%) | 0.078[a] |
|     ≥1 | 223(36.5%) | 124(39.9%) | 99(33.0%) | |
| Margin | | | | |
|     smooth or ill-defined | 470(76.9%) | 239(76.8%) | 231(77.0%) | 0.283[a] |
|     lobulated or irregular | 75(12.3%) | 43(13.8%) | 32(10.7%) | |
|     extrathyroidal extension | 66(10.8%) | 29(9.3%) | 37(12.3%) | |
| Calcifications | | | | |
|     microcalcifications | 196(32.1%) | 123(39.5%) | 73(24.3%) | **< 0.001**[a] |
|     macrocalcifications | 415(67.9%) | 188(60.5%) | 227(75.7%) | |
| Capsular invasion | | | | |
|     positive | 96(15.7%) | 35(11.3%) | 61(20.3%) | **0.002**[a] |
|     negative | 515(84.3%) | 276(88.7%) | 239(79.7%) | |
| TI-RADS | | | | |
|     3 | 8(1.3%) | 4(1.3%) | 4(1.3%) | **0.022**[a] |
|     4a | 176(28.8%) | 104(33.4%) | 72(24.0%) | |
|     4b | 213(34.9%) | 112(36.0%) | 101(33.7%) | |
|     4c | 76(12.4%) | 29(9.3%) | 47(15.7%) | |
|     5 | 21(3.4%) | 7(2.3%) | 14(4.7%) | |
|     6 | 117(19.1%) | 55(17.7%) | 62(20.7%) | |
| Clinically CLNM | | | | |
|     positive | 85(13.9%) | 27(8.7%) | 58(19.3%) | **< 0.001**[a] |
|     negative | 526(86.1%) | 284(91.3%) | 242(80.7%) | |

**Notes.**
[a]The chi-square test was adopted.
[b]The Student's $t$-test was adopted.
Variables with statistical significance are shown in bold.
PTMC, papillary thyroid microcarcinoma; CLNM, central lymph node metastasis; SD, standard deviation; TI-RADS, thyroid imaging reporting and data system.

## US image acquisition and pre-processing

The US images were obtained from the hospital Picture Archiving and Communicating System (PACS). US examination was performed using a US machine (Resona R9 & 7S, Mindray Medical, Shenzhen, China and Acuson Sequoia, Siemens, Erlangen, Germany) equipped with a 2- to 12-MHz linear phased-array transducer. A radiologist, with 20 years of expertise in thyroid ultra-sound, retrospectively chose one representative transverse or longitudinal image. Only the data which successfully met the quality control standards were incorporated. The selected images have been saved as JPG files.

These raw images were cropped manually to contain the rectangular region of interests (ROIs), where the nodes were located in the center by an experienced radiologist using ImageJ software (https://imagej.net/ij/). Each image is initially resized to $64 \times 64$ pixels, followed by the division of each pixel value by 255, prior to inputting each image into the DL model. Additionally, during the training stage, we augment each image by using the function RandomResizedCrop in Pytorch 2.1.0. to random resize and crop each image to $64 \times 64$ pixels.

## Construction of the DCNN model

We used five-fold cross-validation for model training, utilizing 80% of the patients for training and the remaining 20% for validation. The training dataset was applied to establish the model. Meanwhile, the validation set was utilized to assess the effectiveness of our proposed CLNM prediction model. Because patients have images and CFs, we design a deep convolutional neural network (DCNN), including twenty-three convolutional layers and one fully-connected layer with the input dimension being 512, based on ResNet (*He et al., 2016*) to handle images, and we design a multilayer perceptron (MLP), including three fully-connected layers, to handle CFs. We show their detailed architecture of these two models in Fig. 2. Furthermore, similar to a popular strategy (*Wu et al., 2022*), we also present the flowchart to handle both image and CFs in Fig. 3, where only modifies the DCNN model by changing the number of dimensions in fc to 521, which is obtained by adding the number of channels of image feature maps (*e.g.*, 512) and the number of dimensions of CFs (*e.g.*, 9). For both DCNN and MLP models, we implement them with the PyTorch framework, *i.e.*, Pytorch 2.1.0 and Python 3.9 with CUDA 12.2. We adopt the optimizer, Adam, to update the model parameters, with setting the learning rate as 0.0001 and initializing the momentum parameters $\beta_1 = 0.9$ and $\beta_2 = 0.999$. Additionally, we totally train the model 200 epochs and set the batch size to be 64.

## Statistical analysis

IBM SPSS statistics 25.0 software (SPSS, Chicago, IL, USA) and Python 3.6 pack-ages were used in this study. Continuous data are expressed as mean $\pm$ standard, while categorical data is presented by numbers and percentages. To compare categorical variables like gender, we utilized the $\chi 2$ test or Fisher's exact test. On the contrary, we compared continuous variables such as age and tumor diameter utilizing the Student's $t$-test or the Mann–Whitney $U$ test. We designated all statistical significance levels as two sided, with $p < 0.05$ was considered statistical significance. The assessment of performance
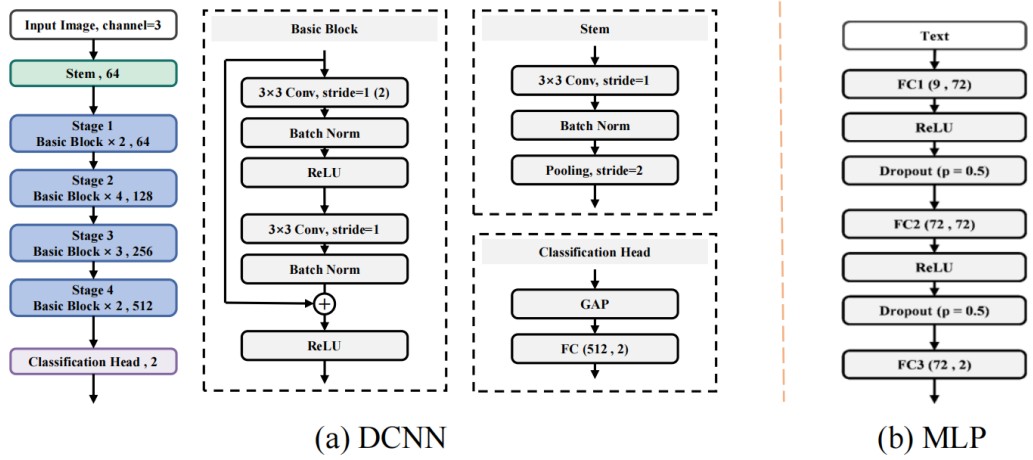

(a) DCNN                                                        (b) MLP

**Figure 2** The detailed architecture of the designed DCNN and MLP for images and CFs, respectively.

relied on the computation of parameters such as accuracy, sensitivity, specificity, positive predictive value (PPV), negative predictive value (NPV), and $F_1$ score. Moreover, the receiver operating characteristic (ROC) curve served to illustrate the effectiveness of the DL model. Subsequently, the area under the curve (AUC) was calculated accordingly.

## RESULTS

### Clinicopathological characteristics of PTMC

The study involved a total of 611 patients composed of 456 females (74.6%) and 155 males (25.4%). Among them, 540 patients (88.4%) were under the age of 55 and 465 patients (76.1%) were categorized as TI-RADS 4. Of all the patients, 85 individuals (13.9%) had clinical CLNM, whereas 300 patients (49.1%) had pathologically confirmed instances of CLNM. Compared to the clinicopathological characteristics of PTMC patients with CLNM and without CLNM, there were significant differences in gender, age, tumor diameter, calcifications, capsular invasion and TI-RADS ($P < 0.05$). On the other hand, there were no significant differences in Hashimoto's thyroiditis, shape, aspect ratio, and margin between the two groups ($P > 0.05$) (Table 1).

### Contribution of the factors to CLNM in PTMC

Logistic regression was used to further examine statistically significant attributes from the univariate analysis in an effort to identify the independent correlation factors in PTMC with CLNM. The analysis verified that age $\geq$55 (OR = 0.309, $p < 0.001$), tumor diameter (OR = 2.551, $p = 0.010$), macrocalcifications (OR = 1.832, $p = 0.002$), and capsular invasion (OR = 1.977, $p = 0.005$) were independently related to CLNM in PTMC (Table 2).

### The performance of the DL model in five-fold cross-validation

We developed three cohort constructed by US images, clinical factors and both of them. Then, we divided the three cohorts into five-fold cross-validation sets. Figure 4 provides

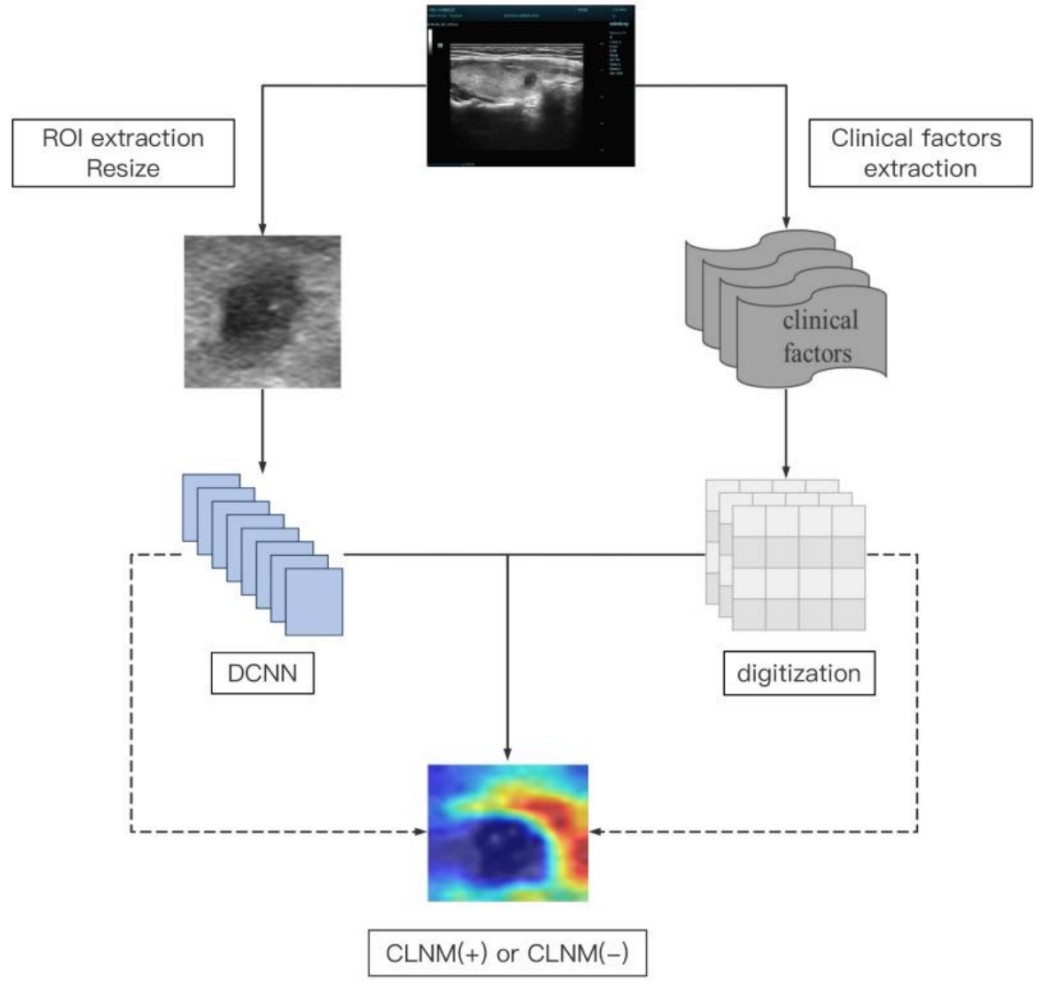

**Figure 3** The flowchart to handle images and CFs, where CFs are represented by digits to add on to the feature map of images.

**Table 2** Binary logistic regression analysis for CLNM in PTMC.

| Characteristics | OR | 95% CI | P |
|---|---|---|---|
| Male | 0.717 | [0.484~1.063] | 0.098 |
| Age ≥55 | 0.309 | [0.174~0.547] | **< 0.001** |
| Tumor diameter | 2.551 | [1.257~5.178] | **0.010** |
| Macrocalcifications | 1.832 | [1.255~2.674] | **0.002** |
| Capsular invasion | 1.977 | [1.230~3.180] | **0.005** |
| TI-RADS 4c | 1.248 | [0.221~7.048] | 0.802 |

**Notes.**

Variables with statistical significance are shown in bold.

PTMC, papillary thyroid microcarcinoma; CLNM, central lymph node metastasis; TI-RADS, thyroid imaging reporting and data system; OR, odds ratio; CI, Confidence interval.

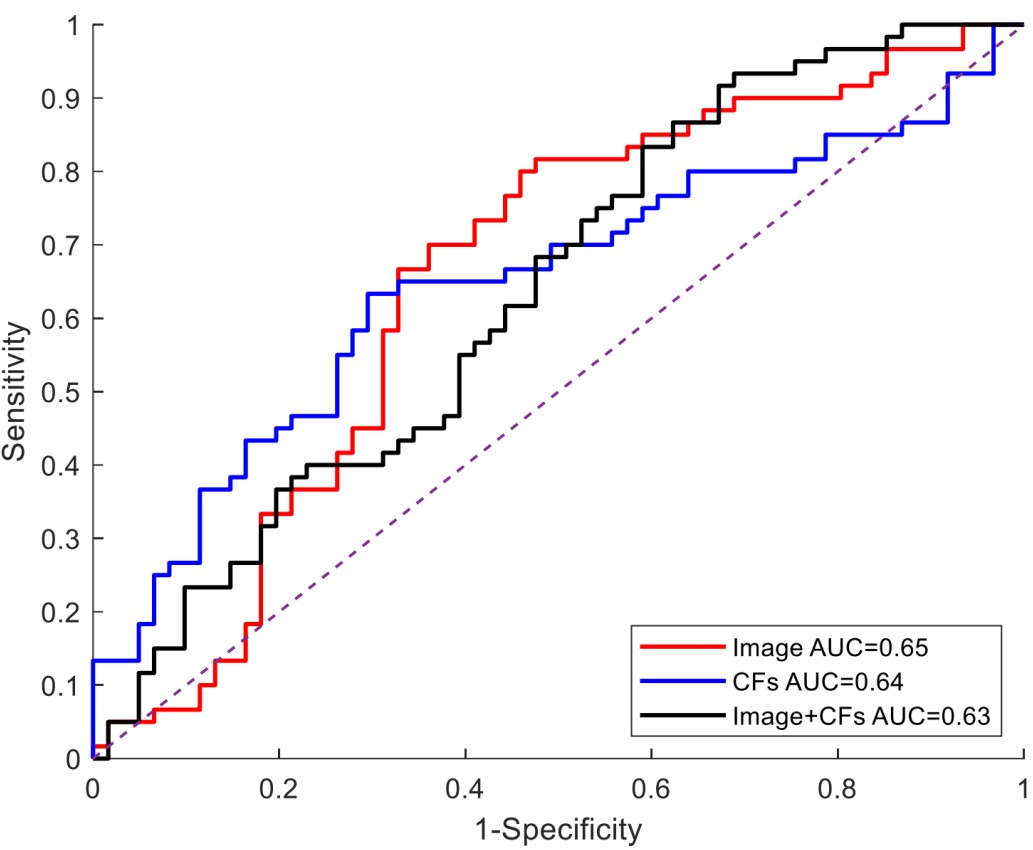

**Figure 4** Comparison of ROC curves in the five-fold cross-validation set by three models. AUC, area under the curve; CFs, clinical factors.

a comparative illustration of the ROC curves for three models based on five-fold cross-validation sets. Experimental results were similar when the model used clinical factors or both US images and clinical factors and the overall AUC is 0.64 and 0.63, respectively. A marginally superior outcome was produced by the model that used US images alone (depicted by the red lines), with an overall AUC of 0.65. A summary of the quantitative indexes for the three different models is depicted in Table 3. Particularly, the DL model based on US images has an ACC value of 65.5%, a sensitivity rate of 71.0%, and a specificity of 56.0%. For the DL model based on clinical factors, the ACC value is 59.7%, the sensitivity rate is 62.0% and the specificity is 55.0%. For the DL model based on both US images and clinical factors, the ACC value is 65.5%, the sensitivity rate is 58.0% and the specificity is 77.0%. Additionally, we apply a popular visualization method Grad-CAM (*Selvaraju et al., 2017*), which weights the 2D activations by the average gradient, to visualize our model. As it is suggested in (*Selvaraju et al., 2017*), we utilize the last convolutional layer of DCNN for visualization, and exhibit the images with model attention in Fig. 4, which displays the ultrasound images of five cases with or without CLNM, respectively, and the visualization

of their corresponding network features. In Fig. 5, the model pays more attention to the area with red color (higher class activation mapping (CAM) weight).

## DISCUSSION

PTMC without clinically evident ETE or LNM is adopted active surveillance strategies in clinical practice (*Sugitani et al., 2021*). Generally, every 3–6 months, people should undergo regular ultrasound examinations to evaluate changes in thyroid nodules. Although PTMC is generally an indolent tumor, LNM will occur in an early stage. The central compartment of the neck is the most frequent location for LNM. To improve the efficiency of PTMC management, it is important to timely and accurately detect the CLNM in PTMC and adopt effective treatment modalities during active surveillance (*Xue et al., 2018*). However, there are anatomic areas of the central region that are not well visualized by ultrasound. Ultrasound has encountered great challenges in the identification of CLNM (*Hwang & Orloff, 2011*). Moreover, the diagnostic accuracy of ultrasound on LNM is severely affected by operator differences. Hence, it is an immediate need to enhance the precision of preoperative prediction of LNM, particularly in the central region.

PTMC without clinically LNM is adopted active surveillance and undergo regular ultrasound examinations every 3–6 months. It is very crucial to timely and accurately detect the LNM in PTMC during the active surveillance process for PTMC management. However, the diagnostic accuracy of traditional ultrasound is relatively low and severely affected by operator differences. To overcome these limitations, several researchers have attempted to predict the LNM in PTC, and they can roughly divided into two categories: the clinical factor-based methods (*Feng et al., 2022a*) and radiomic features-based traditional machine learning (ML) methods (*Wang et al., 2023*). Unfortunately, these methods remain several limitations. The clinical factor-based methods have relatively low accuracy and the conclusions of some studies were inconsistent. Additionally, the highthroughput features extracted in traditional radiomics are easily affected by the imaging parameters, which make them fail to be applied in clinical practice. Therefore, it is an urgently need to design new methods to compensate for the lack of previous research and improve the accuracy of preoperative prediction of cervical LNM, especially in the central region. We developed three DL models using ultrasound (US) images, clinical factors and both of them. Our DL-based diagnostic model can be easily incorporated into existing treatments and assist in the decision-making process. Compared to the previous methods, applying the DL model during a regular ultrasound examination can improve diagnostic efficiency, accuracy as well as reduce clinical workload.

Some univariate and multivariate analyses have shown that ultrasound features include tumor size, thyroid invasion, and microcalcifications are independent indicators of LNM of PTC ($P < 0.05$) (*Gao et al., 2021*; *Guang et al., 2021*). These clinical factors were also validated to be interrelated with CLNM of PTMC in our study. Furthermore, we constructed a DL model to forecast CLNM in PTMC with the aid of US images, clinical indicators, and both of them. Additionally, we use a five-fold cross validation to avoid the impact of low data volumes. The experimental results revealed that the AUCs of the three models on the validation set were respectively 0.65, 0.64, and 0.63. Although the three models

**Table 3**  The quantitative results of three models.

|  | AUC | ACC | SENS | SPEC | PPV | NPV | $F_1$ score |
|---|---|---|---|---|---|---|---|
| Image | 0.65 | 0.66 | 0.71 | 0.56 | 0.62 | 0.67 | 0.66 |
| CFs | 0.64 | 0.60 | 0.62 | 0.55 | 0.61 | 0.70 | 0.60 |
| Image and CFs | 0.63 | 0.66 | 0.58 | 0.77 | 0.68 | 0.64 | 0.61 |

**Notes.**

AUC, area under the curve; ACC, accuracy; SENS, sensitivity; SPEC, specificity; PPV, positive predictive value; NPV, negative predictive value; CFs, clinical factors.

**Figure 5**  Visualization of network features of seven cases with and without CLNM, respectively.

of our study achieve a moderate level of accuracy, it improves quite a lot compared to the traditional ultrasound, and is preliminarily validated to be used for the prediction of CLNM in PTMC. Previous studies (*O'Connell et al., 2013*) have shown that traditional ultrasound can only detect 20–31% of central lymph node metastasis (CLNM), and our study suggests that the accuracy of traditional ultrasound is only 13.9% (Table 1). Additionally, the diagnostic accuracy of traditional ultrasound based on clinical experience is severely affected by operator differences. Comparatively, the deep learning (DL) model, which integrates ultrasound images and clinical factors, overcomes observer variances and has good consistency. Among three models used in this study, the deep model relied generally more on image modalities than the data modality of clinic records when making the predictions. Additionally, the popular strategy to add CFs to the image feature maps cannot always boost the model accuracy. This suggests that it is very necessary to propose more general fusion strategies to leverage the information of images and CFs.

Some limitations existed in our study. Firstly, the size of dataset is relatively insufficient to train a more convincing model. Additionally, our data collection was limited to our single-center, preventing us from extending the verification of our model's robustness. Hence, in the future, we aim to collect more data for model training so as to achieve higher prediction accuracy. Secondly, the DL model constructed in this retrospective study was preliminarily validated to be used for the prediction of central cervical LNM in PTMC. However, affecting by poor quality of ultrasound images, the experimental results are not very satisfying. In order to further boost the model performance, prospective studies also need to be designed to achieve higher prediction accuracy. By prospectively collecting study data, it can collect more high-quality ultrasound images and more comprehensive clinical data.

## CONCLUSIONS

This article designs deep learning models as the reference for the treatment and supervision of PTMC. It suggests that deep learning models can obtain better performance than traditional clinical factor-based statistical methods. Additionally, among the three models used in this study, the deep model with image modalities usually has superior performance over that with clinic records on decision-making.

## ACKNOWLEDGEMENTS

We wish to thank all of the anonymous patients who participated in this study.

### Funding
The authors received no funding for this work.

### Competing Interests
The authors declare there are no competing interests.

### Author Contributions
- Yu Wang conceived and designed the experiments, performed the experiments, analyzed the data, prepared figures and/or tables, authored or reviewed drafts of the article, and approved the final draft.
- Hai-Long Tan conceived and designed the experiments, prepared figures and/or tables, and approved the final draft.
- Sai-Li Duan performed the experiments, prepared figures and/or tables, and approved the final draft.
- Ning Li analyzed the data, authored or reviewed drafts of the article, and approved the final draft.
- Lei Ai analyzed the data, authored or reviewed drafts of the article, and approved the final draft.

● Shi Chang analyzed the data, authored or reviewed drafts of the article, and approved the final draft.

## Human Ethics

The following information was supplied relating to ethical approvals (i.e., approving body and any reference numbers):

Xiangya Hospital Central South University granted Ethical approval to carry out the study within its facilities (Ethical Application Ref: 202211733).

## Data Availability

The raw data are available in the Supplementary Files.

## Supplemental Information

Supplemental information for this article can be found online at http://dx.doi.org/10.7717/peerj.16952#supplemental-information.

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
