# Peer review of "Predicting central cervical lymph node metastasis in papillary thyroid microcarcinoma using deep learning"

_PeerJ, doi:10.7717/peerj.16952_

## Round 0.1 · original submission · Major Revisions

The manuscript titled "Predicting Central Lymph Node Metastasis in Papillary Thyroid Microcarcinoma using Deep Learning" has received valuable feedback from three reviewers. The overall sentiment is positive, with acknowledgment of the innovative approach but with several constructive comments to enhance the study's quality and impact.

**Language Note:** The review process has identified that the English language must be improved. PeerJ can provide language editing services - please contact us at copyediting@peerj.com for pricing (be sure to provide your manuscript number and title). Alternatively, you should make your own arrangements to improve the language quality and provide details in your response letter. – PeerJ Staff

·

Basic reporting

The authors have made a noteworthy attempt to predict central lymph node metastasis (CLNM) in papillary thyroid microcarcinoma (PTMC) using a deep learning (DL) model. The integration of ultrasound (US) images and clinical factors into the DL model is an innovative approach. The study is well-organized, and the methodology is clearly laid out.

1. Model Performance: The model achieves a moderate level of accuracy (AUC of 0.65). The authors should discuss the clinical significance of this level of accuracy, given that the improvement over traditional methods is marginal. This would help in understanding the practical applicability of the model in a clinical setting.
2. Data Size and Diversity: The study uses data from a single center, which might limit the model's generalizability. Expanding the dataset to include multiple centers with diverse patient demographics would potentially enhance the model's robustness and applicability.

Experimental design

3. Comparative Analysis: The study would benefit from a more detailed comparison between the proposed model and existing diagnostic methods. This would include not only performance metrics but also aspects like ease of use, time efficiency, and cost-effectiveness.
4. Model Interpretability: While deep learning models are often considered as 'black boxes', providing some insights into how the model makes its predictions could be valuable, especially in a clinical context where understanding the basis for a decision is crucial.

Validity of the findings

5. Retrospective Nature of the Study: The authors acknowledge the retrospective nature of the study. Future work should aim to validate the model prospectively to assess its real-world effectiveness.

Additional comments

• There are instances of technical terminology that could be better explained for the benefit of non-specialist readers.
• Some minor grammatical corrections are needed to improve the readability of the manuscript.
Overall, this study provides a promising approach towards the use of DL in predicting CLNM in PTMC. Addressing the points above would strengthen the paper and its contribution to the field.

Reviewer 2 ·

Basic reporting

no comment

Experimental design

Line 104 “chosen one representative transverse or longitudinal image” When scanning transverse or longitudinal, we will not get the same nodule with different images, so in your dataset, should you use transverse or longitudinal images consistently?

Validity of the findings

no comment

Additional comments

Line 101 “US image acquisition and pre-processing” Please add the type of ultrasound machine and probe
Line 152 In Result, “Age >55 was independently related to CLNM in PTMC”. While it was not mentioned in the discussion.
Please enrich the discussion section.
Figure 1 Misspelling of the word in the picture “lasion”.

Reviewer 3 ·

Basic reporting

Very limited information on the deep learning model has been provide by the authors.
1. The abbreviation "US" in line 26 should be spelled out.
2. I cannot find model code files in the submitted_codes.zip. The code files are not relevant. Please provide readme files for the codes.
3. The authors do not provide details of the DCNN/MLP models in Table 1. Are there any batch-normalization layers and max-pooling layers? What is the dropout rate? Please provide more details of the model.
4. The author does not provide dependent packages such as pytorch for training the model.

Experimental design

This work does not follow the right machine learning process or the relevant details are not provided.

Generally, to train and evaluate a machine learning model, the dataset is first divided into the training set and the testing set. Then, the training dataset is used for variable selection, hyperparameter selection, and model training. Some of these processes may depend on prediction results, so the training set can be further split into a training set and a validation set. Please note, that the test set should be set aside without being involved in these processes until testing. In this way, the performance will not be overestimated.


In line 113, the authors only mentioned five-fold cross-validation. 80% samples for training and 20% samples for validation. No testing test is defined but a test set is used for testing the performance of the model (line 115). Maybe the author use the validation set as the test set. In this way, the performance will be overestimated. For each fold, there should be training set for training, validation set to decide early stopping and find other hyper-parameters, and testing set to be set aside for testing.

The author also does not provide any detail in visualizing the image and the model. Only the visualization result is provided in Figure 4.

Validity of the findings

Due to the less accurate machine learning process, the result of this work is not rigorous. I suggest the author provide more details on the model and training detail.

---

## Round 0.2 · accepted · Accept

The manuscript has received positive feedback from two reviewers, indicating that the issues raised in the initial reviews have been successfully addressed. The work is commended for its clarity, well-established experimental design, and recognized novelty of the findings. Based on the reviewers' feedback, the manuscript is now in a strong position, and further steps may include minor revisions for enhanced clarity or additional context, if deemed necessary. The positive response from the reviewers suggests that the manuscript is well-prepared for potential publication.

·

Basic reporting

Well Written

Experimental design

Well establish

Validity of the findings

Novelty

Reviewer 3 ·

Basic reporting

The issues have been addressed.

Experimental design

The issues have been addressed.

Validity of the findings

The issues have been addressed.